# Oxidative Stress-Mediated Antibacterial Activity of the Total Flavonoid Extracted from the *Agrimonia pilosa* Ledeb. in Methicillin-Resistant *Staphylococcus*
*aureus* (MRSA)

**DOI:** 10.3390/vetsci9020071

**Published:** 2022-02-07

**Authors:** Liren He, Han Cheng, Fuxin Chen, Suquan Song, Hang Zhang, Weidong Sun, Xiaowei Bao, Haibin Zhang, Chenghua He

**Affiliations:** 1College of Veterinary Medicine, Nanjing Agricultural University, Nanjing 210095, China; 2019107090@njau.edu.cn (L.H.); chenghan@stu.njau.edu.cn (H.C.); suquan.song@njau.edu.cn (S.S.); swd100@njau.edu.cn (W.S.); haibinzh@njau.edu.cn (H.Z.); 2School of Chemistry and Chemical Engineering, Xi’an University of Science and Technology, Xi’an 710054, China; chenfuxin@xust.edu.cn; 3Key Laboratory of Advanced Drug Preparation Technologies, Ministry of Education, Co-Innovation Center of Henan Province for New Drug R & D and Preclinical Safety, School of Pharmaceutical Sciences, Zhengzhou University, Zhengzhou 450001, China; hangzhang@zzu.edu.cn; 4College of Food Science and Pharmacy, Xinjiang Agricultural University, Urumqi 830052, China; xiaoweibao0723@xjau.edu.cn

**Keywords:** oxidative stress, total flavonoid, *Agrimonia pilosa* Ledeb. (*A**. pilosa* Ledeb.), methicillin-resistant *Staphylococcus aureus* (MRSA)

## Abstract

(1) Background: Methicillin-resistant *Staphylococcus* *aureus* (MRSA) is a zoonotic pathogen that causes endocarditis, pneumonia, and skin diseases in humans and livestock. (2) Methods: The antibacterial effect of the total flavonoid against MRSA (ATCC43300) extracted from the *Agrimonia pilosa* Ledeb. (*A**. pilosa* Ledeb) was evaluated by the microdilution method. The oxidative stresses in MRSA were evaluated by the levels of intracellular hydrogen peroxide (H_2_O_2_), reactive oxygen species (ROS), and oxidative stress-related genes. The DNA oxidative damage was tested by the 8-hydroxy-2′-deoxyguanosine (8-OHdG) and DNA gel electrophoresis. The differentially expressed proteins were determined by the method of SDS-PAGE and NanoLC-ESI-MS/MS, while the mRNAs of differential proteins were determined by Real-Time PCR. The changes of ultra-structures in MRSA were observed by Transmission Electron Microscope (TEM). (3) Results: The minimum inhibitory concentration (MIC) of the total flavonoid against MRSA was recorded as 62.5 μg/mL. After treatment with the total flavonoid, the levels of intracellular H_2_O_2_ and ROS were increased and the gene expressions against oxidative stress (*SodA*, *katA*, *TrxB*) were decreased (*p* < 0.01), while the gene expression for oxidative stress (*PerR*) was increased (*p* < 0.01). The level of intracellular 8-OHdG in MRSA was increased (*p* < 0.01) and the DNA was damaged. The results of TEM also showed that the total flavonoid could destroy the ultra-structures in the bacteria. (4) Conclusions: The total flavonoid extracted from the *A**. pilosa* Ledeb can induce the oxidative stress that disturbed the energy metabolism and protein synthesis in MRSA.

## 1. Introduction

*Staphylococcus aureus* (*S. aureus*) is a zoonotic pathogen that can produce staphylococcal enterotoxin (SE) and lead to serious and sometimes fatal infections in humans [1]. In clinic, it was able to cause endocarditis, pneumonia, osteomyelitis, arthritis, and skin diseases in humans [2], as well as cause mastitis in dairy cows and sheep, exudative epidermitis in piglets, and septicemia in young rabbits [3]. In 1960s, a methicillin-resistant *S. aureus* (MRSA) strain was first isolated and reported in England [4]. Later on, MRSA was also isolated from the cows, milker’s nose [5], poultry farms (breeding hens, laying hens, broilers, and turkeys) [6], and pig’s nose [7]. MRSA has become a serious threat to food processor, food samples, and customers [8]. Moreover, MRSA infection in hospitals or communities has been reported in many countries [9]. The treatment of MRSA infection has become a thorny problem in clinic. Therefore, searching for an alternative to treat MRSA infection has become an important research focus. This medical plant extract is a promising alternative, which has many advantages including few side effects, low cost, and less susceptibility to bacterial resistance [10].

*Agrimonia pilosa* Ledeb. (*A**. pilosa* Ledeb) is a rosaceous plant recorded in the Chinese Pharmacopoeia. The decoction of *A**. pilosa* Ledeb has been used to treat gastritis, gastric ulcer, diarrhea, and other diseases in humans and livestock [11]. The bioactive compounds in *A**. pilosa* Ledeb include flavonoid, triterpenoids, and isocoumarin, and have been reported to show many pharmacological effects including anti-inflammatory, antidiabetic, antitumor, anti-acetylcholinesterase, antioxidant, antibacterial, and antiparasitic activities [12,13]. It was reported that an ether extract from the *A**. pilosa* Ledeb had antibacterial activity against *Staphylococcus*
*aureus*, *bacillus,* and *Nocardia* [14]. However, until now, the effect of the total flavonoid against MRSA extracted from *A**. pilosa* Ledeb and the corresponding bacterial mechanism have never been reported. This study aims to evaluate the antibacterial effects of the total flavonoid against MRSA extracted from *A**. pilosa* Ledeb and to elucidate the oxidative stress-mediated antibacterial mechanism further.

## 2. Materials and Methods

### 2.1. Bacterial Strain and Cultivation

The MRSA strain (ATCC43300) was kept in the clinical microbiology laboratory (Nanjing Agricultural University). The strain MRSA was cultured at 37 °C in the Mueller-Hinton broth (MHB) and shaken at 180 rpm.

### 2.2. Preparation of the Total Flavonoid

The medical plant of *A**. pilosa* Ledeb was purchased from Bozhou Traditional Chinese Medicine Market (Bozhou, China) in December 2019 and stored in Nanjing Agricultural University (Voucher no. 20191205). Five hundred grams of dried and powdered *A**. pilosa* Ledeb was sonicated in 3000 mL ethyl acetate for 30 min and kept at room temperature for 24 h. The extract was filtrated and lyophilized under reduced pressure. Next, 50 g of ethyl acetate extract was used to extract the total flavonoid by the method of 60% alcohol reflux [15]. The total time for the alcohol reflux was 1.5 h. The yield of the total flavonoid from the ethyl acetate extract was 6.3% (w/w).

### 2.3. MIC Determination

The microdilution method was used to evaluate the MIC in the 96-well plates. In brief, the total flavonoid was double diluted into a 96-well plate from 2 mg/mL, then the plate was dried at 50 °C for 3 h. After that, 100 µL of MRSA ATCC43300 (1 × 10^6^ CFU/mL) was added into the 96-well plate and the plate was incubated for 12 h at 37 °C. Next, 50 µL of resazurin (0.5 mg/mL) was added into the plate, and the plate was incubated for a further 1.5 h. The discoloration was used to determine the MIC [16]. Finally, the MIC of the total flavonoid against MRSA was recorded as 62.5 µg/mL.

### 2.4. HPLC and HRMS Analysis of the Total Flavonoid

The total flavonoid was detected by high-performance liquid chromatography (HPLC) equipped with a UV detector (Shimadzu Prominence, Japan) and Agilent TC-C18 column (250 mm × 4.6 mm, 5 µm). The mobile phase A was 5% acetonitrile +0.1% formic acid and mobile phase B was 95% acetonitrile +0.1% formic acid. The gradient elution program was 0.01–5 min 30% B, 5–10 min 60% B, 10–40 min 65% B, 40–60 min 100% B, and 60–65 min 0% B. The detection wavelength, flow rate, column temperature, and sampling volume were 254 nm, 0.6 mL/min, 40 °C, and 10 µL, respectively.

The total flavonoid was analyzed by the Q Exactive™ mass spectrometer (Thermo, Waltham, MA, USA), according to the published procedure [17]. In brief, the total flavonoid was dissolved in 95% acetonitrile +0.1% formic acid, and the final concentration was 0.1 mg/mL. Then, 5 µL of sample was analyzed by the following conditions: scan mode, FTMS + p ESI full ms; scanning range, 50–1000 m/z; capillary voltage, 3500 V; dry gas, 8 L/min; temperature, 320 °C. The mass results were analyzed by the software NIST 20 (available online: https://www.sisweb.com/software/ms/nist.htm, accessed on 7 September 2021) including the mass bank library and mainlab database.

### 2.5. Determination of Time–Kill Curves

The total flavonoid was dissolved in 15 mL MHB with the final concentration of 0.25 × MIC, 0.5 × MIC, 1 × MIC, 2 × MIC, and 4 × MIC, respectively. Ceftiofur sodium (10 μg/mL) was used as the positive control (positive group) and the normal MHB was used as the negative control (NC). After adding of the MRSA ATCC43300 strain (1 × 10^6^ CFU/mL), all tubes were cultured at 37 °C with shaking at 180 rpm. Then, the colony was counted every 2 h on the MHB agar plate. The time–kill curves were drawn according to the results of the colony counting.

### 2.6. The Level of Intracellular H_2_O_2_

The levels of intracellular H_2_O_2_ in MRSA ATCC43300 were determined by the H_2_O_2_ Assay Kit (Nanjing Jiancheng Bioengineering Institute, Nanjing, China). The MRSA ATCC43300 was cultured at 37 °C in MHB and shaken at 180 rpm. The concentrations of the total flavonoid in tubes were 0.25 × MIC, 0.5 × MIC, 1 × MIC, 2 × MIC, and 4 × MIC, respectively. Ceftiofur sodium (10 μg/mL) was used as the positive control, and the normal MHB was used as the negative control (NC). After coculture for 8 h, the bacteria were collected by centrifugation (3000 rpm, 15 min, 4 °C) and lysed by ultrasonic waves (200 W, 5 min, 4 °C). The supernatant was used to measure the level of H_2_O_2_ at 405 nm. At the same time, the total proteins in the supernatant were detected by the protein assay kit (Nanjing Jiancheng Bioengineering Institute, Nanjing, China). The H_2_O_2_ and standard protein concentration (163 mmol/L) were provided with the kit. The levels of intracellular H_2_O_2_ were calculated by the following formula:$$H_{2}O_{2}\;(\mathrm{mmol}/\mathrm{mg})=\frac{\mathrm{OD}\;\left(\mathrm{sample}\right)-\mathrm{OD}\;\left(\mathrm{NC}\right)}{\mathrm{OD}\;\left(\mathrm{standard}\right)-\mathrm{OD}\;\left(\mathrm{NC}\right)}\times \frac{163\;\left(\mathrm{mmol}/L\right)}{\mathrm{total}\;\mathrm{protein}\;\mathrm{concentration}\;\mathrm{of}\;\mathrm{the}\;\mathrm{sample}\;\left(\mathrm{mg}/L\right)}$$

### 2.7. The Level of Intracellular ROS Analysis

The levels of intracellular ROS in the 1 × MIC group and NC group were determined by the 2′-7′-dichlorofluorescin diacetate (DCFH-DA) with laser confocal microscope. In brief, after co-incubation for 8 h, the bacterial cells were collected by centrifugation (3000 rpm, 10 min, 4 °C) and washed by PBS for three times, then the fluorescent probe DCFH-DA (10 uM) was added and co-incubated for 2 h at 37 °C with shaking at 50 rpm. After washing with PBS three times, 50 µL of bacterial cell (1 × 10^6^ CFU/mL) was put on the slide, then fixed and observed under the laser confocal microscope. The fluorescence intensity was analyzed by the software of Image J (available online: https://imagej.nih.gov/ij/index.html, accessed on 7 July 2020).

### 2.8. Oxidative Stress-Related Genes Analysis

The Bacterial RNA Extraction Kit (Vazyme Biotech Co., Ltd., Nanjing, China) was used to extract the total RNA in the 1 × MIC group and NC group, respectively. The *16S* gene was used as the internal standard for the following analysis. The primers of the oxidative stress-related genes (*sodA*, *katA*, *TrxB,* and *perR*) and *16S* gene were showed in the Table 1. After reverse transcription, the cDNA, relative primers and SYBR qPCR Master Mix (Vazyme Biotech Co., Ltd., Nanjing, China) were mixed and performed by the StepOnePlus™ Real-Time PCR (Applied Biosystems, CA, USA). The PCR program included an initial denaturation for 30 s at 95 °C, following by 40 cycles of amplification of 95 °C for 5 s and 60 °C for 30 s. The method of 2^−ΔΔ^ CT was used to analyze the data of Real-Time PCR.

### 2.9. 8-Hydroxy-2′-Deoxyguanosine (8-OHdG) and DNA Gel Electrophoresis

The levels of 8-OHdG in the 0.25 × MIC, 0.5 × MIC, 1 × MIC, 2 × MIC, 4 × MIC, positive, and NC group were determined by the 8-OHdG ELISA kit (Jiangsu Meibiao Biotechnology Co., Ltd., Suzhou, China) [18]. The MRSA ATCC43300 was cultured at 37 °C and shaken at 180 rpm in MHB. After co-incubation for 8 h, the bacteria were collected by centrifugation (3000 rpm, 15 min, 4 °C) and lysed by ultrasonic waves (200 W, 5 min, 4 °C). The absorbance of the supernatant at 450 nm was used to measure the level of 8-OHdG. Different concentration of 8-OHdG was used to establish the standard curve, then the concentration of 8-OHdG in the sample was calculated by the standard curve.

The levels of DNA damage in the 0.25 × MIC, 0.5 × MIC, 1 × MIC, 2 × MIC, 4 × MIC, positive, and NC group were evaluated by DNA gel electrophoresis [19]. After co-incubation for 8 h, the bacteria were collected by centrifugation and lysed by the lysozyme, then the total DNA was extracted by the bacterial genomic DNA Extraction Kit (Vazyme Biotech Co., Ltd., Nanjing, China). The extracted DNA was detected by the 1% agarose gel electrophoresis and taken a photo for analysis.

### 2.10. SDS-PAGE and NanoLC-ESI-MS/MS Analysis

The strain MRSA ATCC43300 was cultured at 37 °C and shaken at 180 rpm in MHB which contained 1 × MIC of the total flavonoid. The normal MHB was used as the negative control (NC). After co-incubation for 8 h, the bacteria were collected by centrifugation (3000 rpm, 15 min, 4 °C) and washed with PBS three times. Then, the total proteins were extracted for protein gel electrophoresis. After staining with Coomassie dye, the gel was taken photo and the density of the differential protein bands was analyzed by the software of Image J (available online: https://imagej.nih.gov/ij/index.html, accessed on 7 July 2020).

The differential protein bands were cut and analyzed by the method of NanoLC-ESI-MS/MS [20]. In brief, the cut protein bands were digested with modified trypsin (Promega) and the digested peptides were extracted out with acetonitrile. Then the extracted peptides were dried and redissolved in a sample solution (2% acetonitrile 97.5% water, 0.5% formic acid). The peptides were analyzed by NanoLC-ESI-MS/MS. The data were analyzed by the software PLGS (v 2.3) and searched in the KEGG database (available online: https://www.kegg.jp/, accessed on 6 August 2021).

### 2.11. The mRNA Expression Analysis of Differential Proteins

The primers for the differential protein genes (*crr*, *rplD*, *fba*, *fda*, *pdhB,* and *16S*) were shown in Table 2. The RNA extract and the method of Real-Time PCR were the same as that in the section of oxidative stress-related genes analysis.

### 2.12. Transmission Electron Microscope (TEM) Observation

The bacteria in the 1 × MIC and NC group were observed by the TEM (HT7700, HITACHI), respectively. The strain MRSA ATCC43300 was cultured at 37 °C and shaken at 180 rpm in MHB which contained 1 × MIC of the total flavonoid. The normal MHB was used as the negative control (NC). After cocultivation for 8 h, the bacteria were collected by centrifugation (3000 rpm, 15 min, 4 °C) and washed with PBS three times. Then, the bacterial cells were fixed by 2.5% glutaraldehyde at 4 °C overnight and observed by TEM [21].

### 2.13. Statistical Analysis

All experiments were repeated three times and analyzed by software SPSS 13.0 with the method of One-Way ANOVA. ## means *p*-value < 0.01 and # means *p*-value < 0.05.

## 3. Results

### 3.1. HPLC and HRMS Analysis of the Total Flavonoid Extracted from the A. pilosa Ledeb.

The chromatogram of the total flavonoid extracted from the *A**. pilosa* Ledeb is shown in Figure 1. Compared with the mass spectrum data and literatures, 10 flavonoid compounds were found in the total flavonoid extracted from the *A**. pilosa* Ledeb, including apigenin [22], kaempferol [23], kaempferol-3-*O*-glucoside [24], luteolin [25], quercetin [26], taxifolin [27], tiliroside [28], isoquercetin [29], rutin [27], and vitexin [30] (Appendix A). The total ion chromatogram of the total flavonoid is available in the Appendix A.

### 3.2. The Time–Kill Curves

The time–kill curves of the total flavonoid against MRSA ATCC43300 showed that, at the low concentration of the total flavonoid (0.25 × MIC and 0.5 × MIC), the bacteriostasis on MRSA ATCC43300 was very weak. However, at the high concentration of the total flavonoid (1 × MIC, 2 × MIC, and 4 × MIC), the bacteriostasis was increased. After 10 h, the number of MRSA ATCC43300 decreased to the undetectable level (Figure 2). These results indicate that the total flavonoid extracted from the *A**. pilosa* Ledeb can inhibit the growth of MRSA ATCC43300.

### 3.3. The Level of Intracellular H_2_O_2_

After treatment with the total flavonoid extracted from the *A**. pilosa* Ledeb, the levels of intracellular H_2_O_2_ were significantly increased (*p* < 0.01) compared with the NC group (Figure 3). Moreover, with the increase of the concentration, the level of intracellular H_2_O_2_ in MRSA ATCC43300 also increased. These results suggest that the total flavonoid extracted from the *A**. pilosa* Ledeb can significantly increase the level of intracellular H_2_O_2_ in MRSA ATCC43300.

### 3.4. The Level of Intracellular ROS

Compared with the NC group, the fluorescence intensity under the laser confocal microscope was increased significantly after treatment with the 1 × MIC of total flavonoid (Figure 4, Appendix A). These results show that the total flavonoid extracted from the *A**. pilosa* Ledeb can significantly increase the production of ROS in MRSA ATCC43300.

### 3.5. Oxidative Stress-Related Genes

Compared with the NC group, the mRNA expressions of *SodA*, *katA,* and *TrxB* were decreased significantly in the 1 × MIC group, however, the mRNA expression of *PerR* was increased significantly (*p* < 0.01) (Figure 5). The results show that the total flavonoid extracted from the *A**. pilosa* Ledeb can enhance the level of oxidative stress in MRSA ATCC43300 by regulating the expressions of oxidative stress-related genes.

### 3.6. 8-OHdG and DNA Gel Electrophoresis

The level of 8-OHdG was calculated by the following standard curve:Y = 0.0276X + 0.0245 (R^2^ = 0.9862)Y is the concentration of 8-OHdG (ng/mg); X is the OD value of sample.

The levels of 8-OHdG in MRSA ATCC43300 were increased significantly after treatment with the total flavonoid (*p* < 0.01) (Figure 6). The results of DNA gel electrophoresis showed that the DNA trailing was increased significantly after treatment with the total flavonoid (Figure 7). The results show that the total flavonoid extracted from the *A**. pilosa* Ledeb can induce DNA oxidative damage in the MRSA ATCC43300.

### 3.7. SDS-PAGE and NanoLC-ESI-MS/MS Analysis

Compared with the NC group, the a-band (15–25 kDa), b-band (15–25 kDa), c-band (25–35 kDa), d-band (about 35 kDa), and e-band (35–40 kDa) were significantly down-regulated proteins (Figure 8). The results of NanoLC-ESI-MS/MS showed that the a-band, b-band, c-band, d-band, and e-band were PTS system glucose-specific EIIA component, 50S ribosomal protein L4, Fructose-bisphosphate aldolase, Fructose-bisphosphate aldolase class 1, and Pyruvate dehydrogenase E1 component subunit beta, respectively (Appendix A). The densitometry analysis of differential proteins (a-band, b-band, c-band, d-band, and e-band) is available in the Appendix A. These results suggest that the total flavonoid extracted from the *A**. pilosa* Ledeb can significantly inhibit the protein synthesis in MRSA ATCC43300.

### 3.8. mRNA Expression of Differential Proteins

Compared with the NC group, the mRNA expressions of *crr*, *rplD*, *pdhB,* and *fda* were decreased significantly after the treatment with the 1 × MIC total flavonoid (*p* < 0.01), moreover, the mRNA expressions of *fba* also decreased (*p* < 0.05) (Figure 9). The results of mRNA expression of differential proteins are in accord with the result of SDS-PAGE.

### 3.9. TEM

The ultra-structures of bacterial cells in the NC group were clear (Figure 10a1,a2). However, after treatment with the 1 × MIC of total flavonoid, the nucleoid in bacteria had almost disappeared, and the cell wall was deformed. The “empty” areas and some areas filled with grains were found in the 1 × MIC group (Figure 10b1,b2). The results indicate that the total flavonoid can destroy the ultra-structures in MRSA ATCC43300.

## 4. Discussion

MRSA is an important zoonotic pathogen in the field of veterinary and public health which has infected globally in hospitals and communities [31]. It becomes more and more difficult to use antibiotics to treat the MRSA infection in clinic, therefore, an alternative to treat MRSA is needed [32]. The extract of *Canarium odontophyllum* Miq. [33] and the methanol extract of grape seed [34] were reported to have the activity of anti- MRSA. The extract of *A**. pilosa* Ledeb was also reported to have the anti-bacterial activity against MRSA (MIC: 0.1–0.78 mg/mL) [35]; however, the antibacterial mechanism of the extract of *A**. pilosa* Ledeb has never been reported.

Oxidative stress has become a promising antibacterial strategy that focused on the production of reactive oxygen species (ROS) in the bacteria in order to disturb the balance between oxidation and antioxidation [36]. In this work, after treatment with the total flavonoid, the levels of ROS and H_2_O_2_ in MRSA were increased significantly. *SodA*, *katA*, *TrxB,* and *PerR* are the key regulatory genes for oxidative stress in MRSA. The *SodA*, *katA,* and *TrxB* are the coding gene of Mn-superoxide dismutase, catalase, and thioredoxin reductase, respectively, which are responsible for the ROS scavenging in the MRSA. However, the *PerR* is the coding gene of the peroxide resistance regulator which is sensitive to H_2_O_2_ [37]. Our results indicated that the oxidative stress was inspired after treatment with the total flavonoid. 8-OHdG is a product of oxidized nucleoside of DNA induced by oxidative stress and is a reliable marker of oxidative stress [38]. In this present work, the generation of 8-OHdG was increased significantly, suggesting that the DNA in MRSA ATCC43300 was damaged by the ROS and H_2_O_2_ induced by the total flavonoid.

Glucose is the most important carbon source in bacteria. The metabolism of glucose is regulated by phosphoenolpyruvate: glucose phosphotransferase system (PTS) [39]. PTS system glucose-specific EIIA component is a transport system of carbohydrates in bacteria, which is a carbohydrate-specific protein composed of three domains (EIIA, EIIB, EIIC) [40]. Pyruvate dehydrogenase complex (PDC) is one of the key enzymes involved in glucose metabolism under the aerobic condition. It catalyzes the conversion of pyruvate to acetyl coenzyme A, which is an important part of the metabolic energy pathway of organisms [41]. Pyruvate dehydrogenase E1 β is one of the important catalytic enzymes [42], encoded by the *pdhB* gene, which is responsible for the oxidative decarboxylation and energy metabolism of pyruvate. The fructose-bisphosphate aldolase, also called aldolase, is a glycolytic enzyme that catalyzes the conversion of fructose 1-6-diphosphate to glyceraldehyde 3-phosphate (G3P) and dihydroxy-acetone phosphate (DHAP) via the glycolysis metabolic pathway [43]. It also catalyzes the reversible aldol condensation of DHAP with G3P in gluconeogenesis and Calvin cycle. Fructose-bisphosphate aldolase class 1 utilizes an active-site lysine residue to form a Schiff base with the substrate as part of the reaction mechanism. This protein is involved in step 4 of the sub-pathway that synthesizes D-glyceraldehyde 3-phosphate and glycerophosphate from D-glucose [44]. 50S ribosomal protein L4 is one of the main rRNA binding proteins [45]. The protein initially binds to the 5′ end of 23S rRNA. It has multiple contacts with assembled 50S subunits and different domains of 23S rRNA in the ribosome [46]. The decreased mRNA expression of 50S ribosomal protein L4 indicates that the ribosome function is inhibited and the efficiency of protein synthesis is also reduced. Our results showed that after treatment with the total flavonoid extracted from the *A**. pilosa* Ledeb, the expressions of proteins and relevant mRNAs were decreased. It was speculated that the total flavonoid could inhibit the glycolysis pathway and affected energy metabolism.

## 5. Conclusions

The total flavonoid extracted from the *A**. pilosa* Ledeb could inhibit the growth of MRSA ATCC43300, and the MIC was 62.5 μg/mL. Its antibacterial mechanism was oxidative stress induced by the total flavonoid extracted from the *A**. pilosa* Ledeb that disturbed the energy metabolism and protein synthesis in MRSA.

## Figures and Tables

**Figure 1 vetsci-09-00071-f001:**
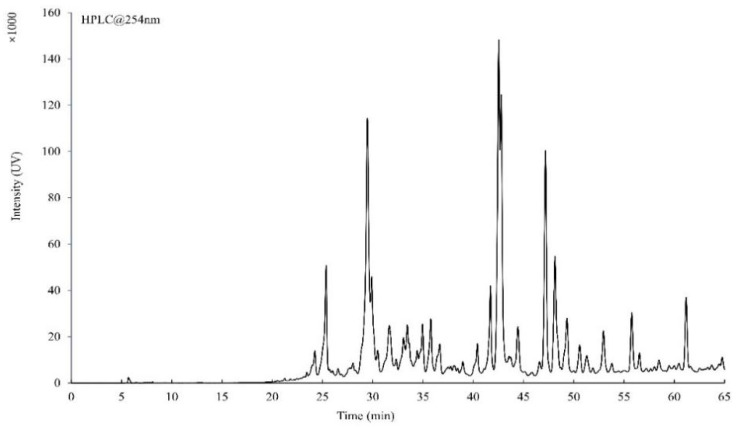
The chromatogram of the total flavonoid extracted from the *A**. pilosa* Ledeb at 254 nm.

**Figure 2 vetsci-09-00071-f002:**
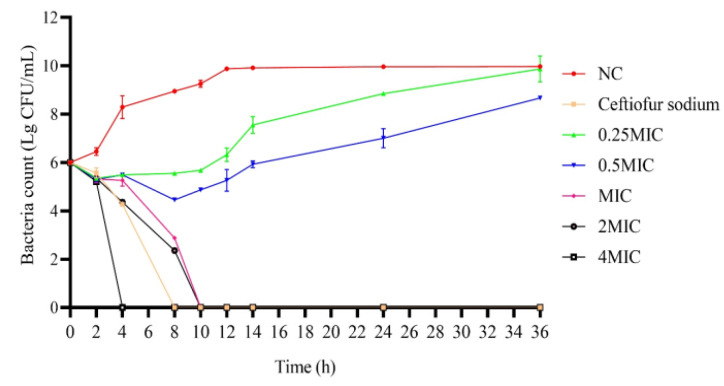
The time–kill curves for the total flavonoid against MRSA ATCC43300 at different concentrations (4 × MIC, 2 × MIC, 1 × MIC, 0.5 × MIC, and 0.25 × MIC). Ceftiofur sodium was used as the positive control. The normal MHB was set as the negative control (NC).

**Figure 3 vetsci-09-00071-f003:**
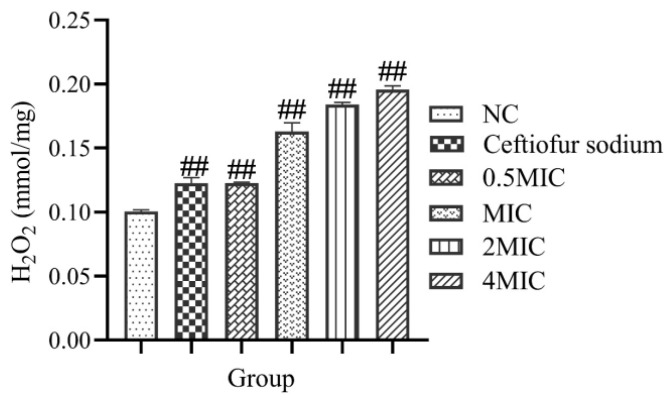
The levels of intracellular H_2_O_2_ in MRSA ATCC43300 after the treatment with different concentrations of the total flavonoid extracted from *A**. pilosa* Ledeb. The Ceftiofur sodium was used as positive control. The normal MHB was set as the negative control (NC). ## means *p*-value < 0.01.

**Figure 4 vetsci-09-00071-f004:**
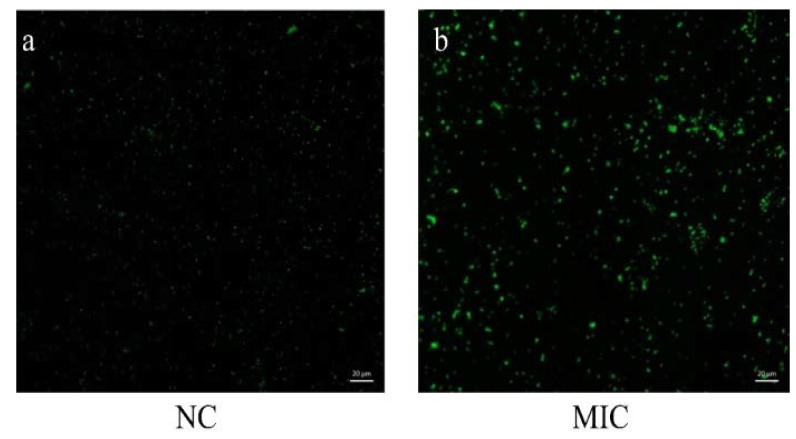
The levels of intracellular ROS in MRSA ATCC43300 after treatment of the total flavonoid extracted from the *A**. pilosa* Ledeb. The normal MHB was set as the negative control (NC). (**a**) was the NC group, (**b**) was the 1 × MIC group.

**Figure 5 vetsci-09-00071-f005:**
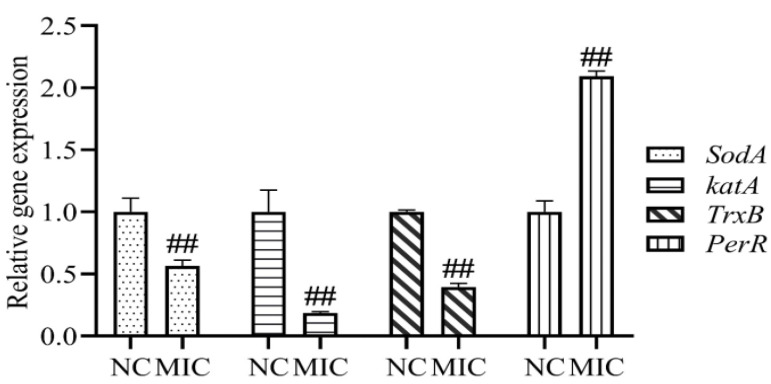
The relative gene expressions of *SodA*, *katA*, *TrxB,* and *PerR* after treatment of the total flavonoid extracted from the *A**. pilosa* Ledeb. The normal MHB was set as the negative control (NC). ## means *p*-value < 0.01.

**Figure 6 vetsci-09-00071-f006:**
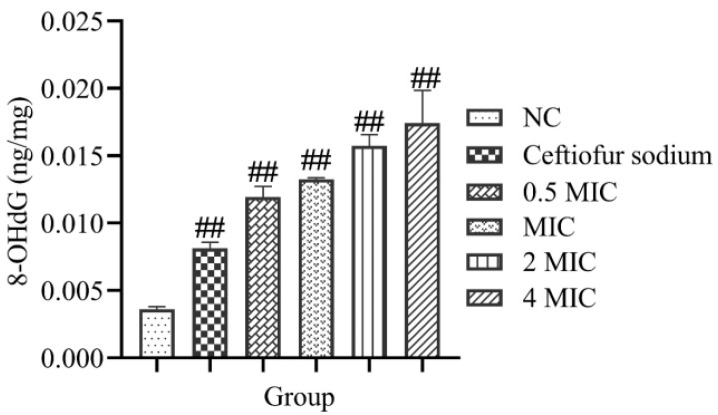
The levels of 8-OHdG in MRSA ATCC43300 after treatment of the total flavonoid extracted from the *A**. pilosa* Ledeb. The normal MHB was set as the negative control (NC). ## means *p*-value < 0.01.

**Figure 7 vetsci-09-00071-f007:**
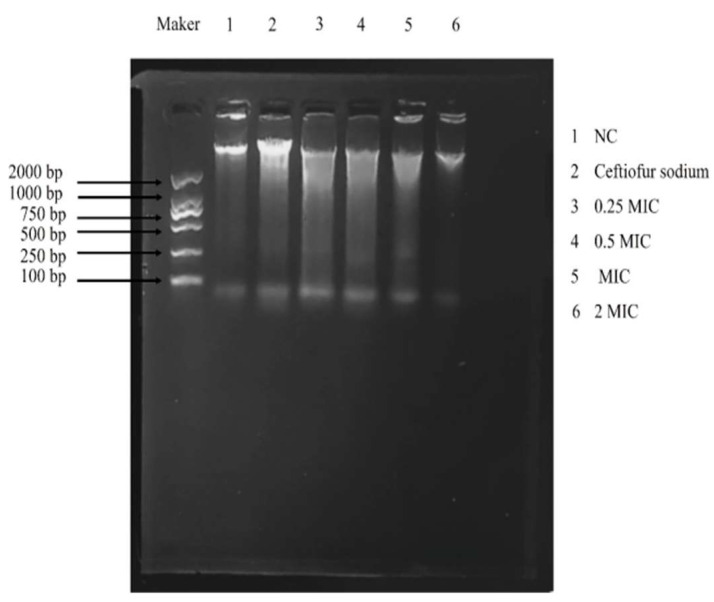
The DNA gel electrophoresis after treatment of the total flavonoid extracted from the *A**. pilosa* Ledeb. The normal MHB was set as the negative control (NC).

**Figure 8 vetsci-09-00071-f008:**
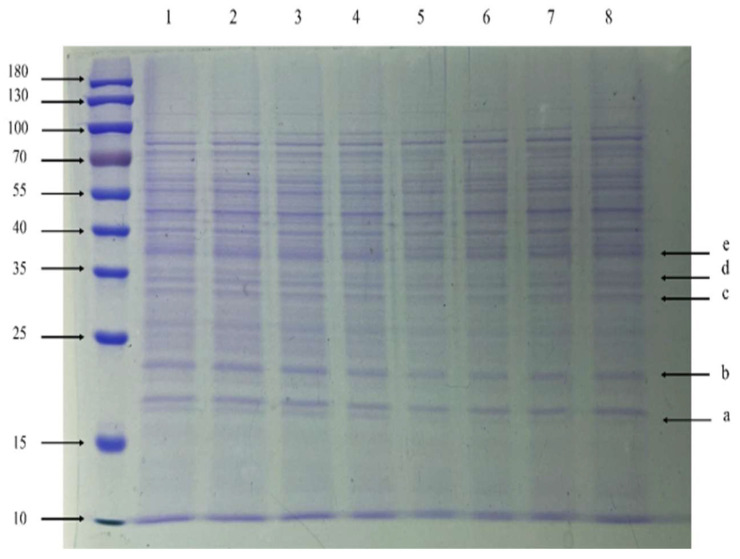
The SDS-PAGE of the total proteins from MRSA ATCC43300 after treatment of the total flavonoid extracted from the *A**. pilosa* Ledeb. Lanes 1–4 were the proteins from the NC group and Lanes 5–8 were the proteins from the 1 × MIC group. Labels (**a**–**e**) indicated the differentially expressed protein bands selected for NanoLC-ESI-MS/MS analysis.

**Figure 9 vetsci-09-00071-f009:**
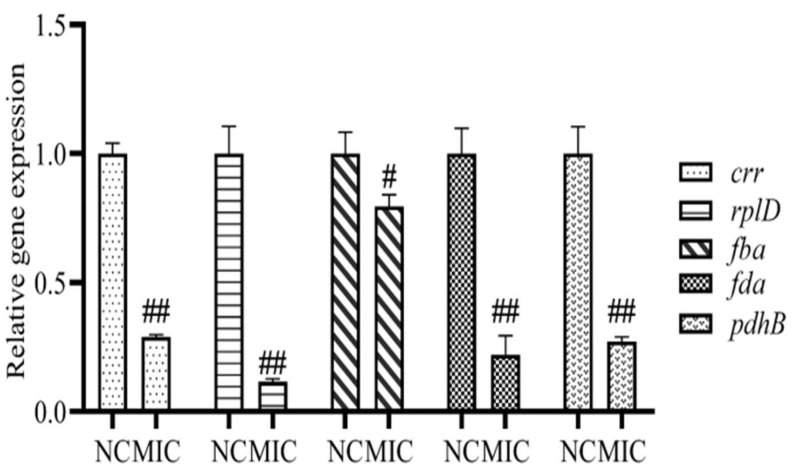
The mRNA expressions of *crr*, *rplD*, *pdhB*, *fba,* and *fda* after treatment of the total flavonoid extracted from the *A**. pilosa* Ledeb. The normal MHB was set as the negative control (NC). ## means *p*-value < 0.01 and # means *p*-value < 0.05.

**Figure 10 vetsci-09-00071-f010:**
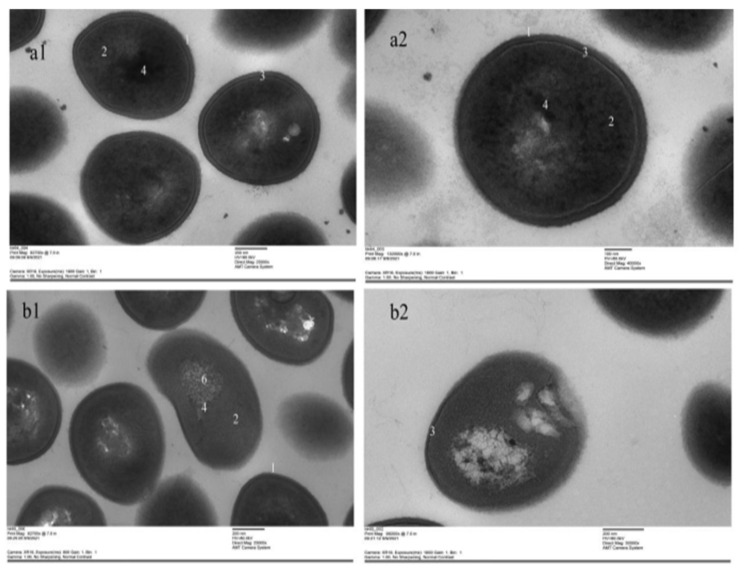
The ultra-structural changes of MRSA ATCC43300 were observed by TEM. (**a1**,**a2**) were the NC group. (**b1**,**b2**) were the 1 × MIC group. 1—cell wall; 2—cytoplasm; 3—intermediate layer; 4—nucleoid; 5— “empty” areas; 6—areas filled with grains. ((**a1**,**b1**), 25,000×; (**a2**), 40,000×; (**b2**) 30,000×).

**Table 1 vetsci-09-00071-t001:** The primers of oxidative stress related genes (*sodA*, *katA*, *TrxB,* and *perR*).

Name	Forward Primer (5′→3′)	Reverse Primer (5′→3′)
*16s*	CCAGCAGCCGCGGTAAT	CGCGCTTTACGCCCAATA
*sodA*	AAGCGTGTTCCCATACGTCTAAACC	TTGGTTCAGGTTGGGCTTGGTTAG
*katA*	GCTGCTGAAATTATAGCTACAGAT	TACTTGAATATACATTGTCCATTT
*TrxB*	AAGACGGCAAAGTGGGTTCTGTG	TGGCGCTGTTAATGGCTTCATACC
*perR*	TCCATTCGATGATGTGTTACGTCA	TGTGAACAATGTGGTAAGATCGTTG

**Table 2 vetsci-09-00071-t002:** The primers for the gene of *crr*, *rplD*, *fba*, *fda,* and *pdhB.*

Name	Forward Primer (5′→3′)	Reverse Primer (5′→3′)
*crr*	TCCTTCGCCGTCTAATTGAACTGTG	GCAGGACGTGTTGACAATGTCTTTC
*rplD*	ACTTCTTGGAGTTGGTCCGAATACG	GGAACAGGTCGTGCTCGTCAAG
*fba*	ACTGAACCTAATGCTGGCGCTAATG	ACTGTTGGCGGACAAGAAGATGATG
*fda*	TGTTCCAACTTGTTCACGATATGCG	GCAGTGTATTTGCCTTCTACTTCGC
*pdhB*	TTGATGCGATTGCTGGACAAATTGC	TGTGTGTACGCCACCACCAAATG
*16s*	CCAGCAGCCGCGGTAAT	CGCGCTTTACGCCCAATA

## Data Availability

Data are available on request from the corresponding author.

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
