# Peer review of "Oxidative Stress-Mediated Antibacterial Activity of the Total Flavonoid Extracted from the Agrimonia pilosa Ledeb. in Methicillin-Resistant Staphylococcusaureus (MRSA)"

_vetsci, 2022, doi:10.3390/vetsci9020071_

Round 1
Reviewer 1 Report
The scientific message of the research is interesting and the manuscript is mostly well organized; however, the authors are advised to revise the manuscript and improve overall writing. Furthermore, to my understanding, the integration under the scope of Veterinary Sciences should be improved.
Specific comments:
- Throughout the manuscript, the usage of italics for genus and species should be ensured. For example, in L.59 the naming of species should be corrected, with the use of italics for the whole name and not only the genus (‘Staphylococcus aureus’), and usage of the first capital letter on the genus (Bacillus).
- References section should be revised, using italics when necessary. It is also advised that the authors reduce self-citations to the minimum.
- I believe the manuscript would gain from language corrections, and when possible, the wording should be more adequate for scientific writing. As an example, L.65-66 would be clearer as: “The strain MRSA was cultured on Mueller-Hinton broth (MHB) at 37ºC with continuous shaking at 180 rpm.” It is also more scientifically correct to use ‘staining’ and not ‘dyeing’ in L.164.
- Another example where the authors could improve clarity and fluidity is the L.92-93, which could be rephrased to: “The total flavonoid was analyzed by the Q Exactive™ mass spectrometer (Thermo, USA), as described by Ma et al. [13].”
- The authors should make sure that they always write “H2O2” (subscript) and not “H2O2”, even in headings and subheadings.
- To facilitate the reader to understand the methodology, the authors could include the formula to calculate the levels of intracellular H2O2 as a formula and not continuous text. Currently, I am unsure if what the authors mean is:
H2O2 (mmol/mg) = (ODsample-ODNC)/(ODstandard-ODNC) ×(163 mmol/L)/([total protein]sample (mg/L))
- 224 there are two full spots.
Major comment:
My main issue with the study is the integration of the research under the scope of Veterinary Sciences, which from my understanding, refers to research relevant to any field of veterinary sciences, including prevention, diagnosis and treatment of disease, disorder and injury in animals.
I believe this would fall under the “treatment” scope, however reading the manuscript the only mention of potential interest to Veterinary Sciences readers is:
“MRSA was an important zoonotic pathogen in the field of veterinary and public health which was infected globally in hospitals and communities”.
I apologize if I am missing other key points, but I believe the authors need to further discuss the implications of such discovery in Veterinary Sciences.
Author Response
Thank you very much for your comments and suggestions. The response to reviewer has been uploaded in the attachment.

Reviewer 2 Report
The paper deals with the antibacterial effects of the total flavonoid extracted from the Agrimonia pilosa Ledeb plant against MRSA, with the aim to elucidate its antibacterial mechanism. The authors are commended for their extensive experimental work and a valid research conclusion. However, the paper is written in very poor English language and syntax and is not readable as such. I recommend that the authors conduct a thorough English editing by a native speaker before resubmitting. Only then can a proper review of the scientific content be conducted.
Author Response

(The authors gave the same response as above.)

Reviewer 3 Report
Dear Authors,
I believe that you have carried out a very interesting study and that it will find a great response from Scholars.
Currently, the problem of AMR has a growing weight in the scientific community and above all in not only human but also animal health. Therefore, given the difficulty or lack of interest in the synthesis of new molecules with antibacterial action, it is essential that researchers turn their attention to the natural molecules deriving from the plant world, intensifying their studies in this way.
In my opinion, the manuscript only needs minor changes to be published, which I report below.
Comments and suggestions:
- line 55: The reference [2] is not consistent with what is reported in the text, please check and replace.
- lines 60-61: Although it is clear from what is described in the title and in the introduction, it would be appropriate to clearly state the aim of the study. Please integrate.
- line 67 - Subsection 2.2: Please specify that the extraction process took place in line with that reported by the authors from another plant species.
Author Response

(The authors gave the same response as above.)

Round 2
Reviewer 1 Report
Thank you for addressing the reviewers comments and concerns.
The changes resulted in a clearer and richer manuscript, particularly the ones made in the introduction, where the association to Veterinary Sciences is now of greater emphasis.
Author Response
Thank you very much for your comments and suggestions.

Reviewer 2 Report
The authors somewhat corrected the text. However, they are still mixing present tense and past tense. The paper really needs to be edited by a native speaker before it is suitable for further review.
For example: Staphylococcus aureus (S. aureus) WAS a zoonotic pathogen; Agrimonia pilosa Ledeb (A. Ledeb) WAS a rosaceous plant; The result of SDS-PAGE WAS shown in Figure 8 etc....
The authors did a solid experimental work and wrote a decent paper. They need to upgrade the basic quality of the language.
Author Response

(The authors gave the same response as above.)
